



# Realistic modelling of faults in LoopStructural 1.0

Lachlan Grose[1], Laurent Ailleres[1], Gautier Laurent[2], Guillaume Caumon[4], Mark Jessell[3], and Robin Armit[1]

[1]School of Earth, Atmosphere and Environment, Monash University PO Box 28E, Victoria, Australia
[2]Université d'Orléans, CNRS, BRGM, ISTO, UMR 7327, Orleans France.
[3]The University of Western Australia, Centre for Exploration Targeting, School of Earth and Environment, Perth, Western Australia, Australia
[4]Université de Lorraine, CNRS, GeoRessources, F- 54000 Nancy, France

**Correspondence:** Lachlan Grose (lachlan.grose@monash.edu)

**Abstract.** Without properly accounting for both fault kinematics and faulted surface observations, it is challenging to create 3D geological models of faulted geological units that are seen in all tectonic settings. Geometries where multiple faults interact, where the faulted surface geometry significantly deviate from a flat plane and where the geological interfaces are poorly characterised by sparse data sets are particular challenges. There are two existing approaches for incorporating faults

into geological surface modelling: one approach incorporates the fault displacement into the surface description but does not incorporate fault kinematics and in most cases will produce geologically unexpected results such as shrinking intrusions, fold hinges without offset and layer thickness growth in flat oblique faults. Another approach builds a continuous surface without faulting and then applies a kinematic fault operator to the continuous surface to create the displacement. Both approaches have their strengths, however neither approach can capture the interaction of faults within complicated fault networks e.g fault du-

plexes, flower structures and listric faults because they either (1) impose an incorrect (not defined by data) fault slip direction; or (2) require an over sampled data set that describes the faulted surface location. In this study we integrate the fault kinematics into the implicit surface by using the fault kinematic model to restore observations and the model domain prior to interpolating the faulted surface. This approach can build models that are consistent with observations of the faulted surface and fault kinematics. Integrating fault kinematics directly into the implicit surface description allows for complex fault stratigraphy and

fault-fault interactions to be modelled. Our approaches show significant improvement in capturing faulted surface geometries especially where the intersection angle between the faulted surface geometry and the fault surface varies (e.g. intrusions, fold series) and when modelling interacting faults (fault duplex).

## 1 Introduction

Building 3D geological models of complex geometries (e.g. fault duplexes, flower structures, folds, fold interference patterns) from only field observations is still a challenge for most 3D modelling approaches. This challenge exists because many of the





observations and structural geology rules are not directly integrated into the mathematical description of the geological models. Structural geologists typically use their knowledge, expertise and bias to digitally reproduce a working hypothesis which often derives from an internally visualised model (Jessell et al., 2014). The modelled geometries will always converge towards the

geologists bias and conceptual model, meaning that model building cannot be used to test different geological hypotheses. An alternative would be to have structural geology concepts encoded in the modelling process. This would enable testing of different geological settings and the geologists' expertise would be used to assess the viability of these different geometries. This strategy has been developed for modelling folded geometries, based on the structural observations and concepts (Laurent et al., 2016; Grose et al., 2017, 2018, 2019), a mathematical description of the fold geometry that combines a fold structural

frame and fold shape profiles. The fold modelling framework has been applied to complex synthetic examples (Laurent et al., 2016; Grose et al., 2017, 2018) and to natural fold examples (Grose et al., 2017, 2019). In this contribution we adapt the fault frame and a kinematic parameterisation of fault geometries (Jessell and Valenta, 1996; Godefroy et al., 2018; Laurent et al., 2013) and apply it directly to the implicit description of the faulted surfaces. Rather than applying the fault kinematics to the modelled surfaces we restore the observations and model domain and interpolate the surfaces without the faults.

Defining the geometry of faults and fault networks is a significant component of understanding and characterising the structural architecture of geological terranes. Geometrical descriptions of fault networks and stratigraphic interfaces are used to refine and improve resource characterisation (Cox et al., 1991; Mueller et al., 1988; Vollgger et al., 2015), geophysical inversions (Blaikie et al., 2014) and for understanding complex geological structures (Putz et al., 2006). Implicit 3D geological modelling methods do not directly incorporate fault kinematics into the surface description and as a result cannot reliably model

the interaction between faults in fault networks, *e.g.* duplex faults, flower structures, relay ramps and listric faults (Cherpeau and Caumon, 2015; Calcagno et al., 2008; de la Varga and Wellmann, 2016). It is challenging to build models that are consistent with both geological observations and fault kinematics because the existing methods require the user to choose whether to use observations of the faulted surface or honour fault kinematic observations/knowledge (Laurent et al., 2013; Godefroy et al., 2018)

Modelling the location of the fault surface is generally approached in a manner analogous to modelling bedding or any other structural feature (Calcagno et al., 2008). The challenge for modelling faults is in characterising how the fault interacts with other structural features, *e.g.* how stratigraphy or other geological features are offset and deformed by the fault. Conceptually there are two approaches for modelling faults: (1) incorporating the fault displacement geometries into the description of the geological surfaces; or (2) apply fault displacements after interpolating a continuous surface.

To incorporate faults into the description of the geological surface a step function can be added to the implicit function (Calcagno et al., 2008; de la Varga and Wellmann, 2016; Marechal, 1984). The step function applies a jump in the scalar field value across the fault discontinuity, however unless (1) the angle between the fault surface and the faulted surface is constant and; (2) the angle between the faulted surface and the fault slip direction is constant the modelled fault is not representative of the true fault movement. There is no direct method of incorporating fault kinematics (movement direction and displacement

magnitude) into the description of the geological surface using step functions. One approach that has been used for incorporating fault displacements is to generate multiple model realisations framing the geological modelling problem as a Bayesian





inverse problem. In this probabilistic framework model data points become parameters for the forward model and fault displacement can be incorporated by including a likelihood function where the fault displacement is expected to come from a prior probability distribution, that may be guided by observations of the fault displacement (de la Varga and Wellmann, 2016; Wellmann et al., 2017; Godefroy et al., 2018). However, this approach is still limited by the inability for fault kinematics to be incorporated into the step function.

The alternative approach is to apply the fault displacement to an already interpolated surface. A first possibility is to apply restoration of the geometric structures surrounding the faults and find the displacement field based on some mechanical likelihood criterion (Thibaut et al., 1996; Maerten and Maerten, 2015). However, finding an objective mechanical criterion is difficult and computationally challenging, an alternative is to construct the geometry of the continuous surface prior to faulting and then to apply the fault operator to the faulted surface (Godefroy et al., 2018; Laurent et al., 2013; Georgsen et al., 2012). In this approach, both the slip direction and the magnitude are used to constrain the fault displacement. Since a continuous surface needs to be interpolated prior to applying the fault, the surface observations within the fault ellipsoid are discarded when interpolating the initial surface. The observations within the fault ellipsoid can then be used to determine the fault displacement magnitude by minimising the misfit between the surface and observations.

Using step functions in the implicit surface description and adding faults to existing surfaces are both useful in basin settings where the pre-fault geometry is reasonably simple and most geometrical complexities are associated with the fault geometry. However, it is unclear how these approaches can be applied to fault networks with a large number of interconnected discrete faults or when modelling complex poly-deformed geometries. When modelling multiple interconnected faults (e.g. splay faults, crosscutting faults and abutting faults), it can be challenging to properly account for the relative kinematics between faults, especially where the observations of the faulted surfaces are limited to a map view.

In this contribution, we adapt the approach presented by Laurent et al. (2013) and Godefroy et al. (2018), by incorporating the fault operator directly into the implicit function. This uses the fault kinematics to restore the model domain and data points to a pre-fault geometry. The restoration function is defined within a structural frame (Grose et al., 2020) where the coordinates define fault surface, fault slip direction and the fault extent. This restoration function allows for the kinematics of the fault to be directly incorporated into the modelling work flow. Faults are modelled starting with the most recent and the kinematics are added to every consecutive geological feature added to the model. This application of time aware geological modelling follows on from previous work for modelling folds by (Grose et al., 2017; Laurent et al., 2016). In this paper we outline the framework for integrating faults used in LoopStructural (Grose et al., 2020). We show case studies ranging in complexity from a simple faulted intrusion where we demonstrate the limitation of step functions, a thrust duplex system, and a faulted fold series using the fold modelling framework from Laurent et al. (2016) and Grose et al. (2017).



## 2 Background 3D modelling methods

### 2.1 Surface representation

Three dimensional structural geological models are a representation of subsurface geology where the geological units are
either represented using boundary surfaces (upper and lower contacts) (Wellmann and Caumon, 2018) or a discretisation on a
predefined support. There are two approaches that are commonly used for representing surfaces in 3D geological models. The
first approach uses explicit surface representation: the geometry of surfaces are contained using a support that is collocated with
the surface geometry. The surfaces are represented using discrete objects such as triangulated surfaces, two-dimensional grids
or parametric surfaces. The surface geometry is usually built by either directly triangulating data points or using interpolation
algorithms to create a smooth surface fitting the data (Chilès and Delfiner, 1999; de Kemp, 1999; Mallet, 1992). The explicit
surface representation means that the geometry of the surface is only represented where the surface is located.

The second approach, implicit surface representation, uses isovalues of one or several scalar fields in 3D space to represent
the geometry of geological surfaces (e.g. stratigraphic horizons and fault surfaces) (Jessell, 1981; Calcagno et al., 2008; Cau-
mon et al., 2013; Cowan et al., 2003; Frank et al., 2007; Hillier et al., 2014; Lajaunie et al., 1997; Mallet, 2002, 2014; Maxelon
et al., 2009; Moyen et al., 2004; Yang et al., 2019; Renaudeau et al., 2019; Manchuk and Deutsch, 2019; Gonçalves et al., 2017;
De La Varga et al., 2019). The value of the scalar field represents the distance away from a reference horizon. Alternatively, if
the geological interfaces are conformable, the surfaces can be represented by isovalues of the scalar field representing the rela-
tive thicknesses between the interfaces. The gradient of the scalar field is a representation of the orientation of the surface being
modelled. These scalar fields can be constructed using a variety of different interpolation methods: e.g. co-kriging (Calcagno
et al., 2008; De La Varga et al., 2019; Gonçalves et al., 2017; Lajaunie et al., 1997), radial basis functions (Cowan et al., 2003;
Hillier et al., 2014) or using discrete interpolation on a predefined support (Irakarama et al., 2020; Caumon et al., 2013; Frank
et al., 2007; Moyen et al., 2004; Mallet, 1992). Most modelling approaches use the same interpolation algorithms for all types
of geological surfaces even though the physical properties and expected geometries can vary.

These interpolation methods mean models can be created with less bias and are more reproducible. Implicit methods
also make it possible to generate a set of model realisations reflecting geological uncertainty (Manchuk and Deutsch, 2019;
Cherpeau et al., 2010; Henrion et al., 2010; Wellmann et al., 2010; Lindsay et al., 2012; Yang et al., 2019).

### 2.2 Faults in implicit modelling

There are a number of existing techniques available for incorporating faults into the implicit approach with methods allowing
for topology (Calcagno et al., 2008; Moyen et al., 2004), kinematics (Laurent et al., 2013; Godefroy et al., 2018) and uncertainty
(Cherpeau and Caumon, 2015; Cherpeau et al., 2012) to be incorporated into the modelling workflow. Incorporating faults into
implicit surface representation is difficult because faults represent a discontinuity in geological feature being modelled and
require a discontinuity in the implicit function. The geometry of the faulted geological feature (stratigraphic interfaces, faults
or foliations) should be consistent with the kinematics of the fault.





There are three possible approaches that can be used to incorporate faults into implicit surface description: (1) interpolate
fault domains using independent implicit functions; (2) incorporate the fault into the domain discretization or; (3) apply a fault
operator to an already interpolated surface.

The first approach separates the faulted feature using two implicit functions, one on the hanging wall and another on the
foot wall. A boolean operation using the fault surface geometry is used to define which implicit function is used to represent
the faulted feature (Caumon et al., 2013; Cowan et al., 2003; Frank et al., 2007). When the hanging wall and foot wall are
represented using separate implicit functions, the displacement of the fault is not explicitly defined and will be a result of
the observations of the faulted feature. There is no incorporation of the fault kinematics unless these are captured by the
location/geometry of observations of the faulted surface. Using different implicit functions for the hanging wall and foot wall
is suitable for areas where the faults cut across the whole model and there is significant displacement and no continuity expected
across the model. For these approaches to work there needs to be a significant amount of data near the fault damage zone to
correctly characterise the fault related deformation, otherwise the kinematic indicators of the layers may not fit the geologists
knowledge about the fault.

The second approach directly incorporates the location of the fault into the polynomial trend of the implicit function using
a discontinuous step function (Calcagno et al., 2008; De La Varga et al., 2019; Marechal, 1984). The fault displacement
magnitude is not used as an observation for building the model but is found by solving the interpolation system for the co-
kriging matrix. de la Varga and Wellmann (2016) applied Bayesian inference to incorporate the fault displacement into the
implicit modelling work flow by using the observations as parameters for the inverse problem. In their approach models with
different displacements are generated by perturbing the observations of the faulted surface. The objective function that is
incorporated into their Bayesian model as a likelihood, only considers the magnitude of the displacement and not the direction
of displacement. This approach provides an interesting idea of using Bayesian inference as a method for optimising the model
parameters in order to try and generate models which fit both geological knowledge and geological observations. An alternative
to this approach for discrete implicit methods is to introduce a discontinuity in the regularisation term, by either cutting the
interpolation support by the fault geometry (Caumon et al., 2013; Frank et al., 2007) or introducing ghost nodes into the discrete
elements (Irakarama et al., 2020).

The third approach integrates the fault kinematics into implicit 3D modelling work flows by using a numerical operator that
defines the distribution of fault slip on and close to the fault surface (Godefroy et al., 2017, 2018; Laurent et al., 2013). In these
approaches a fault operator models fault displacement within an ellipsoidal fault volume. The displacement is discontinuous
across the fault surface and decreases smoothly away from the fault. In this approach, a fault volume is defined where the
fault displacement is non-zero. The geological interfaces are defined by first interpolating a continuous surface across the
fault. Displacement profiles can be identified, either manually defined or using numerical optimisation techniques (Godefroy
et al., 2018). This approach applies the fault to a surface and is not integrated with the implicit function meaning it would not
directly integrate with approaches where the implicit function represents a continuous features such as foliations, conformable
stratigraphy or grade shells.





Alternative kinematic models for describing fault displacements exist such as the trishear model (Erslev, 1991; All-mendinger, 1998). Trishear is a kinematic model describing the geometry of fault propagation folds and has been presented

in pseudo 3D (Cristallini and Allmendinger, 2001) and in true 3D (Cardozo, 2008). Trishear has been used to understand the structural evolution that has resulting in the geometries observed in the field and as part of seismic interpretations. In the interpretation of reflection seismic data, kinematic fault restoration approaches based on vector fields have been proposed (Hale, 2013; Wu et al., 2015). For example, Wu et al. (2015) determine the fault tip by seismic image processing, then interpolate fault shifts in the volume by least-squares smoothing. However, this model is most directly applicable in contexts

where observations are sparse, and the smoothing assumes that fault shifts depend on the stratigraphic orientation. It also restores all faults at once whereas sequential restorations allow for the fault topology to be includes.

## 2.3 Limitations of the presented approaches

None of the presented approaches are capable of incorporating both fault kinematics and geological observations into the the geological surface description. The step function approach is appealing because the fault is incorporated into the model

description, however it cannot capture the kinematics of the fault where the angle of intersection between the fault and faulted surface is variable (e.g. fold series, or intrusions). For example, in Fig. 1A and Fig. 1C a fold series and intrusion are faulted by a reverse fault. In both of these examples the intersection angle between the fault surface and the geometry of the faulted surfaces varies significantly within the model. In Fig. 1B and Fig. 1D the same structures are shown where the fault is incorporated using step functions. Both examples show an increase in the value of the scalar field across the fault geometry, however the gradient

of the implicit function is continuous across the fault. This results in faulted surfaces that do not show the expected kinematics, for example the fold axial traces are not displaced by the fault and the volume of the intrusion shrinks across the fault.

## 3   A kinematic framework for modelling faults

Faults are described by geologists by the movement of rock packages and not only by the location of the discontinuity. Struc-tural geologists describe faults with the fault slip and the displacement magnitude. In a lot of cases measuring the fault slip

direction is impossible, the geologist will have a conceptual model to describe the fault displacement direction, for example using their conceptual model based on geological knowledge. This means that to properly capture the displacement kinematics in structural models the additional kinematic constraints need to be encoded into the modelling algorithms.

In this contribution, the work flows presented by Laurent et al. (2013) and Godefroy et al. (2018) are modified by incorpo-rating the fault operator into the implicit representation of the geological feature in the model (Fig. 2). The faulted stratigraphy

can then be interpolated as a continuous layer where the data points are restored using the fault kinematics. The fault kinematics are integrated into the implicit surface description, meaning that the scalar field when evaluated in the model domain captures the fault kinematics. Integrating the fault kinematics into the implicit modelling system is achieved by:

1. Building the *fault frame*, a curvilinear coordinate system representing the fault geometry
2. Defining the *fault displacement* within the model domain



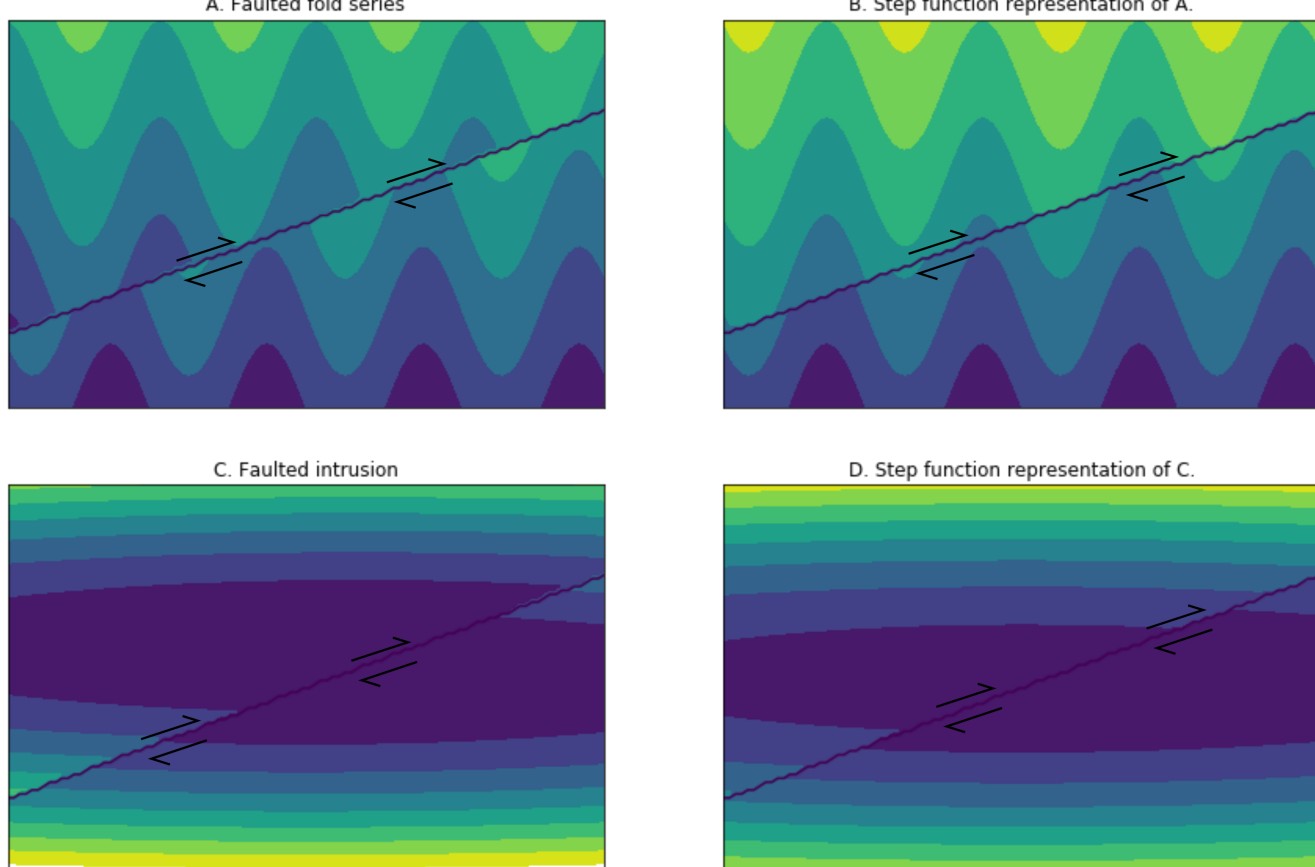

**Figure 1.** Comparison of true fault kinematics and step function results: A) a fold series that has been faulted by a reverse fault, where the fold hinges are offset. B) shows the same fold series as A) when modelled using a step function where the scalar field on the hanging wall is increased resulting in no displacement of the fold hinges. C) is a faulted intrusion where the volume is not affected by the fault. D) shows the same intrusion modelled with a step function where the intrusion shrinks on the hanging wall resulting in a lower volume intrusion.





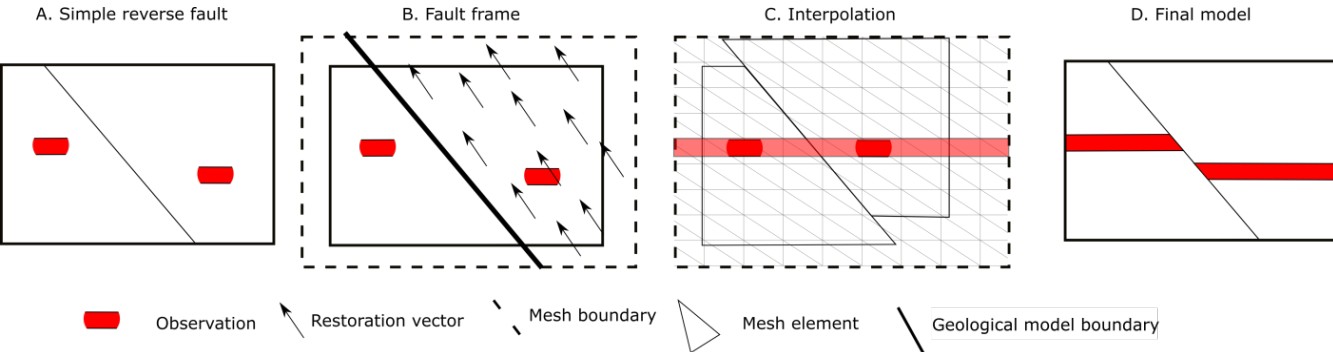

**Figure 2.** Schematic diagram showing application of fault kinematics to implicit modelling. A. observations of reverse fault. B. Fault frame with restoration vectors showing. C. Observations and model domain are restored for interpolation, the mesh is where interpolation occurs is not deformed and is built in the restored space. D. Final model is evaluated by applying fault restoration kinematics and evaluating scalar field from mesh in C.

## 3. Adding the fault kinematics to the implicit surface description

### 3.1 Fault frame

A curvilinear coordinate system (the fault frame, described empirically for natural faults by Walsh and Watterson (1987)) represent different geometrical elements of the fault geometry and is built by interpolating structural observations. The first coordinate ($f_0$) represents the distance to the fault surface and the isovalue of 0 will represent the fault surface and can be interpolated from any structural observations of the fault surface *e.g.* strike and dip controls and observations of the fault surface location. The second coordinate ($f_1$) measures the displacement direction of the fault and the normal to the gradient of this field will be parallel to any kinematic indicators for the fault (*e.g.* slickenslides, stretching lineations) and parallel to the fault surface. The third coordinate ($f_2$) measures the distance in the direction of the fault extent and the gradient of this field will be orthogonal to the gradient norm of the fault surface ($f_0$) and also orthogonal to the fault displacement field ($f_1$).

Three local direction vectors are implicitly defined by the normalised gradient of the fault frame for any location:

$$\mathbf{f_0} = \nabla f_0 / \|f_0\|$$
$$\mathbf{f_1} = \nabla f_1 / \|f_1\|$$
$$\mathbf{f_2} = \nabla f_2 / \|f_2\|$$

$$(1)$$

The fault frame can then be queried for any location within the model returning the distance to the fault centre and the fault frame vectors.





## 3.2 Fault displacement

The displacement direction of the fault is represented by $f_1$. The fault displacement magnitude can be either defined by a constant displacement on the hanging wall or by a function of the fault frame coordinates allowing for the fault displacement to be represented relative to a fault origin using a combination of attenuation profiles as described by Laurent et al. (2013) and Godefroy et al. (2018). We define the local fault displacement vector for any location as

$$f_{disp} = d \cdot f_1 \tag{2}$$

where $d$ is the scalar value of the fault displacement magnitude at that location.

## 3.3 Time aware geological modelling

The aim of implicit 3D geological modelling is to define a function or series of functions that can be evaluated for any location in the model and return some information about the geological feature of interest (either the distance to a feature *e.g.* lithological contact or fault surface or the orientation of the feature). This means that the resulting geological model can effectively be 215 represented by an ordered list of mathematical functions. The topological relationship between these mathematical functions should be related back to the geological topology, *e.g.* the overprinting relationships between geological events.

To represent a faulted geological feature we first define the geological feature in the model using the associated structural data. A fault can then be applied to this geological feature which produces a faulted geological feature. The fault restores the observations attached to the faulted geological feature to their location using the fault kinematics from the fault operator. 220 The faulted geological feature can then be interpolated using the restored locations. To evaluate the scalar field of the faulted geological feature the fault transformation must be added to the implicit surface representation. This is done by applying the fault restoration kinematics to the data points prior to evaluating the scalar field values.

$$\phi_{faulted}(x,y,z) = \phi_{base}\left( f_{disp}(x,y,z) \right) \tag{3}$$

Using this work flow it is possible to stack multiple fault operators and apply, backwards in time, the appropriate displacements 225 to restore the location to its position prior to faulting. This approach can be applied to any structural elements in the model, including lineations, fold vergence, tectonic foliations and fold frames (Grose et al., 2017).

## 4 Implementation

The generalised work flow presented above does not depend on the specifics of the interpolation scheme. We present an overview of how to build the fault frame using discrete interpolation algorithms using LoopStructural (Grose et al., 2020), 230 a open source 3D modelling library[1]. Using discrete interpolation algorithms, the complexity of the interpolation problem is defined by the resolution of the support and not by the number of constraints.

---

[1]https://github.com/Loop3D/LoopStructural





## 4.1 Building fault frame

The fault frame is built by interpolating the scalar fields using the following procedure:

1. Interpolate coordinate 0 to represent the geometry of the fault surface so that the isosurface of 0 contains the fault trace and the field is parallel to the orientation of the fault surface.

2. Interpolate coordinate 1 so that the direction of it's gradient norm is orthogonal to direction of the gradient norm of fault surface and parallel to any kinematic indicators for the fault.

3. Interpolate coordinate 2 so that the direction of its gradient norm is orthogonal to the direction of the gradient norm of the fault surface field and to the fault slip direction field.

In this study we use a discrete implicit modelling technique where the implicit function is represented on a discrete support by either a piece-wise linear function on a tetrahedral mesh (Mallet, 2002; Frank et al., 2007; Caumon et al., 2013; Grose et al., 2020) or by using finite differences on a Cartesian grid where an objective function such as the bending energy (Renaudeau et al., 2019) or the second derivative is minimised (Irakarama et al., 2020). To build the fault frame, an additional constraint needs to be applied to the interpolation to ensure that the fault slip direction and fault extent fields are orthogonal. To add this additional constraint the gradient of an existing component of the fault frame ($\nabla f_i$) (for example the gradient of the fault surface) is obtained for every element in the mesh and this gradient norm is then used to constrain the interpolation of a new field ($\nabla f_j$) so that the gradient norm of the new field is orthogonal to the gradient norm of the existing field.

$$\nabla f_i \cdot \nabla f_j = 0 \tag{4}$$

The scalar field can then be interpolated by minimising the misfit between the orthogonality constraint and any observations of the fault slip direction.

The element orthogonality constraint can be applied to two fields and the resulting scalar field will be close to orthogonal to both. Frequently it is difficult to measure or observe the fault slip direction. In these cases the geologists would need to use qualitative geological knowledge to define the fault slip direction using their knowledge about the geology of the area. It is also possible to build the model with different fault slip directions (and magnitudes) in order to explore the range in possible geometries and compare these with observations.

## 4.2 Volumetric fault displacement

Fault slip is defined by three functions applied to the normalised fault frame coordinates. Where the fault is active the fault frame is coordinates will be between -1 and 1 and be equal to -1 or 1 where the fault is inactive. The same approach for combining the fault profiles within the fault frame is used to define a volumetric fault displacement field as Godefroy et al. (2018), which was adapted from Laurent et al. (2013):

$$\delta(X) = D_0(f_0(X)) \cdot D_1(f_1(X)) \cdot D_2(f_2(X)) \tag{5}$$





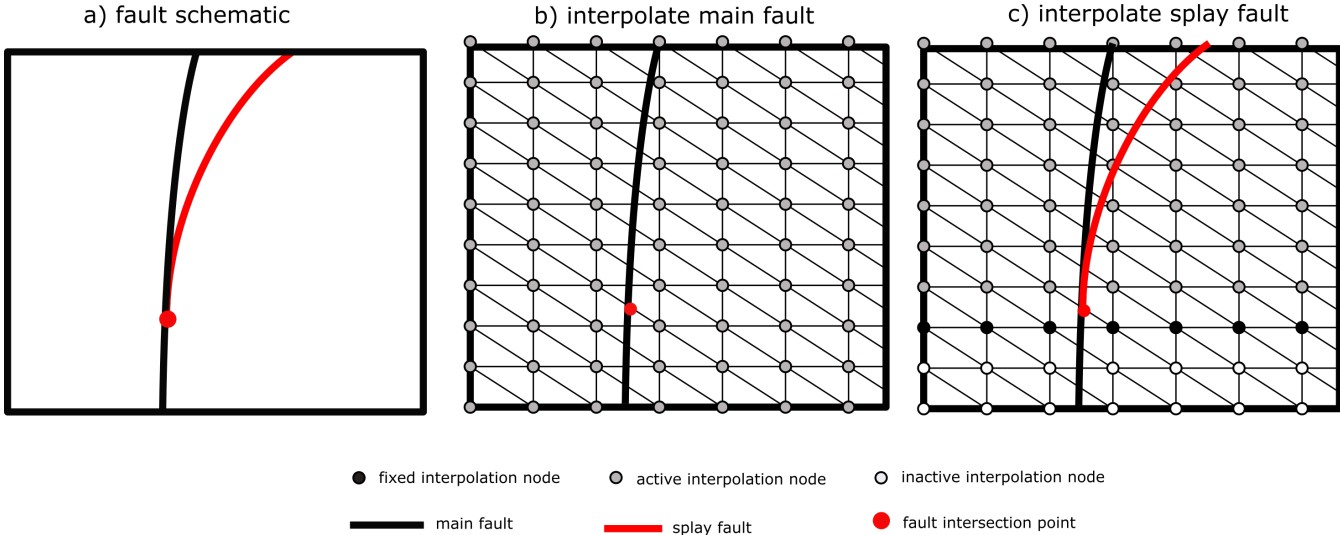

**Figure 3.** Schematic diagram showing interpolation of splay faults. a) fault diagram showing the geometry of the main fault, splay fault and point of intersection. b) Interpolation support for main fault showing fault geometry and all interpolation nodes being active. c) interpolation of splay fault showing fixed, active and inactive interpolation nodes where the fault geometry is shared.

Where $D_{0,1,2}$ are 1-D curves describing the displacement of the fault within the fault frame.

A unit vector representing the displacement direction ($e_d$) can be locally obtained using the fault frame coordinates. We use the local direction vector $e_d(X) \cdot \delta(X)$ to define the local slip vector for the fault. This scaled displacement field is calculated for the entire model support and can then be queried for the local model of interest.

### 4.3 Splay faults

Faults are rarely single surfaces and are often defined as fault zones with complex linkage between individual fault segments. During the development of a fault zone, an individual surface can splay meaning that the displacement of the fault is separated onto multiple slip surfaces (Huggins et al., 1995; Marchal et al., 2003). The resulting geometries can be increasingly complex and includes duplexes and flower structures.

An advantage of our approach, is that multiple faults can be applied sequentially where the total displacement is defined by the accumulation of the local displacement fields. Modelling faults in a time aware approach is similar to how folds are modelled (Laurent et al., 2016; Grose et al., 2017, 2018) where the most recent fold is modelled first and this defines the geometry of the folded foliations.

A fault zone can be represented by a piece-wise combination of fault segments. To model multiple connected faults the fault frames need to equal along the branch line. In the following section, hard constraints are incorporated into the interpolation system using using Lagrange multipliers.





Discrete interpolation approaches define the interpolation within a volume of interest by a linear system where the unknowns are the property values on the mesh nodes (Frank et al., 2007; Caumon et al., 2013). The unknowns are found by solving the

280 linear system:

$$A \cdot x = b \qquad (6)$$

Where $A$ is an interpolation matrix that contains the different geological constraints (contact observations, orientation constraints, orthogonality constraints) and a regularisation term that controls the smoothness of the resulting scalar field and $x$ are the values of the scalar field at the interpolation nodes. The linear system defined in equation 6 is over determined and can be

solved in a least squares sense by minimising the L-2 norm $\|Ax - b\|_2^2$.

Splay faults (Fig. 3a) share the same geometry as the parent fault at the point of intersection. To ensure the faults share the same geometry an additional constraint needs to be added to the linear system so that the scalar field value of the shared nodes of the fault are equal. In Fig. 3b the main fault is interpolated using all interpolation nodes. To interpolate the splay faults the hard constraints need to be identified by finding the values of the fault extent field ($f_2$) where the fault is active. Equality

constraints are identified by finding the elements that contain the isosurface of the fault extent field containing the intersection, point e.g., black nodes in Fig. 3c. The hard constraints form an an additional linear system,

$$C \cdot x = d \qquad (7)$$

where $C$ is the interpolation matrix for the equality constraints.

To solve this system ensuring that the hard constraints are strictly honoured and the other interpolation constraints are

295 satisfied in a least squares sense we define the Lagrangian for the least squares problem,

$$\mathcal{L}(x, \lambda) := \frac{1}{2}\|Ax - b\|_2^2 + \lambda^T (Cx - d), \qquad (8)$$

where $\lambda$ are the Lagrange multipliers. To apply this constraint we need to know where the two faults share geometries so that the values of the main fault can be used as equality constraints to interpolate the splay fault. Where the geometry of the two surfaces is shared, the property value of the nodes inside the shared region are fixed so that both surfaces have the same value

(Fig. 3c). This can be implemented where $C_{i,j} = 1$ where $i$ is index of hard constraint and $j$ is the node global index. $d_j$ is equal to the shared node value between the two scalar fields. Alternatively, the hard constraints could be defined using the discrete support shape functions (*e.g.* linear tetrahedron or trilinear interpolation), however in this case the line of intersection would need to be known, which may be complicated or impossible to quantify.

In Fig. 4B and C, two splay faults are interpolated, in B) the same constraints are used to solve both surfaces and the resulting

faults overlap but do not share the same geometry in the region where they are expected to splay. In B) equality constraints are used to constrain the surface geometry to be the same before the fault splay. When modelling complex fault networks for example splay faults or other faults with shared geometries the hard constraints ensures that the faults have the same geometry at intersection lines. For splay faults this will also ensure that the interpolation of the splay fault also includes the parent fault geometry.





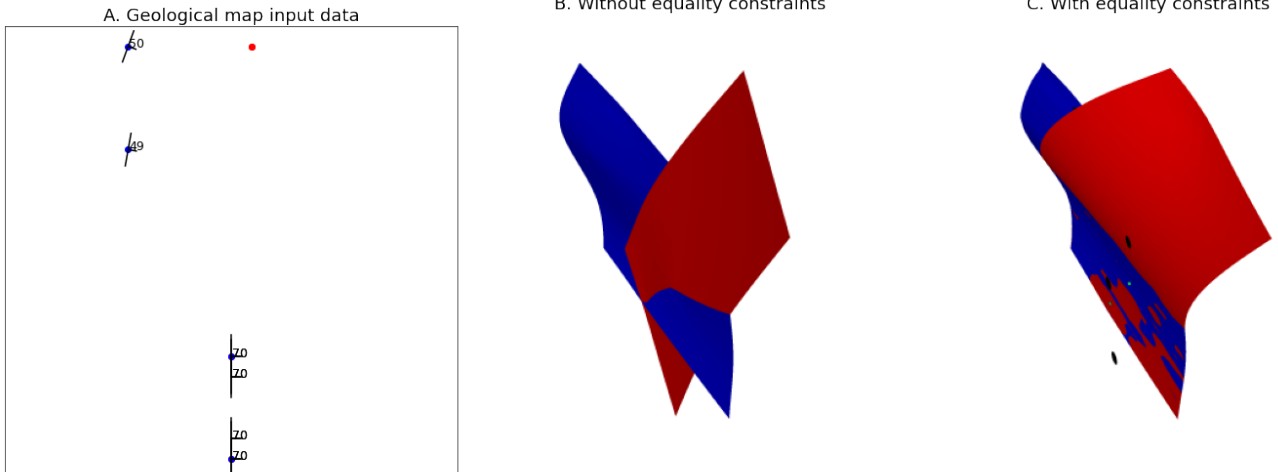

**Figure 4.** Comparison of fault surfaces modelled using equality constraints and without equality constraints. A) input datasets showing observations for both faults. B) Fault surfaces modelled without equality constraints. C) fault surfaces modelled with equality constraints.

## 5 Case studies

In the following sections we will present three synthetic case studies that demonstrate the application of our approach to both extensional and contractional systems. All of the case studies have been implemented in the open source 3D structual modelling library, LoopStructural (Grose et al., 2020). The case studies are provided as interactive Jupyter notebooks in Appendix A.

### 5.1 Faulted intrusion

The first case study is a synthetic intrusive body that has been faulted by a planar normal fault. The intrusion has been randomly sampled with one set of data points describing the distance to the intrusion surface, where the value is 0 and another set of data points where the value is 1 which describes the distance to the intrusion surface. In Fig. 5A the observations of the faulted surface are shown along with the interpolated fault surface. The fault displacement is pure dip slip with a normal sense, meaning the points on the hanging wall (left side) have been displaced down. A vector field representing the fault restoration is defined by multiplying the fault slip direction field (orthogonal to the fault surface and parallel to observations of the fault slip) by the fault displacement magnitude. In this example a fault displacement of $500m$ is used. Fig. 5B shows the fault displacement vector field applied to the data points on the hanging wall of the fault indicating the direction of movement to restore the data points to their location prior to faulting. The data points that are identified as being on the footwall of the fault are not affected by the fault restoration and are grey coloured without a vector field. Fig. 5C shows the data points from Fig. 5A in the restored locations after applying the fault operator. The intrusion shape is interpolated within the restored space Fig. 5C,D using a discrete interpolation approach with a weak regularisation constraint to allow for the propagation of a high curvature geometry. A weak regularisation constraint is applied because modelling intrusions with implicit interpolation approaches has similar



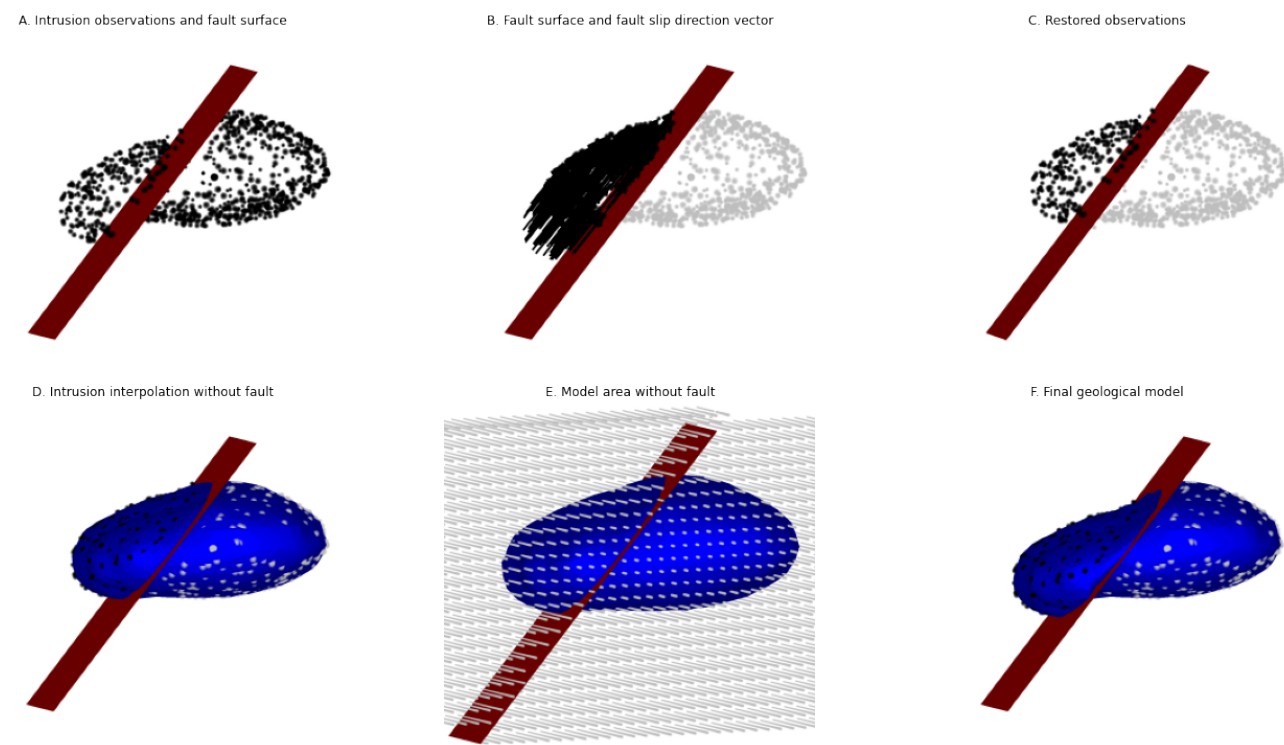

**Figure 5.** Case study showing faulted 3D intrusion: A) observations of the intrusion surface (black points) and modelled fault surface (red surface). B) fault restoration vector evaluated on data point locations showing the restoration direction. C) restored data points from A. D) intrusion surface interpolated in restored domain showing data points and fault surface. E) regular grid highlighting restoration function on model domain, where the hanging wall points have been restored. F) fault and faulted intrusion surfaces

challenges to modelling folds (Laurent et al., 2016), where high curvature is expected but the interpolation actually minimises the curvature. A kinematic operator is added to the interpolated scalar field so that when evaluating the scalar field at a location, the location is restored using the fault displacement field. Fig. 5E shows the model support for a regular grid where the points in the hanging wall are restored using the restoration vector field. In practice, the points are not actually moved and the restoration vector field is actually added to the implicit surface description resulting in a faulted intrusion Fig. 5F.

In this example the intrusion volume is maintained when the fault is applied. The volume of an intrusion when modelled using step functions is likely to change as shown in Fig. 1.

## 5.2 Finite fault

In the previous examples we have shown a constant displacement on the hanging wall of the faults. Applying a constant displacement makes the assumption that the fault displacement is uniform across the fault and that the faults are infinite throughout the model area. This assumption of uniformity, is useful where the faults extend across the model domain. In





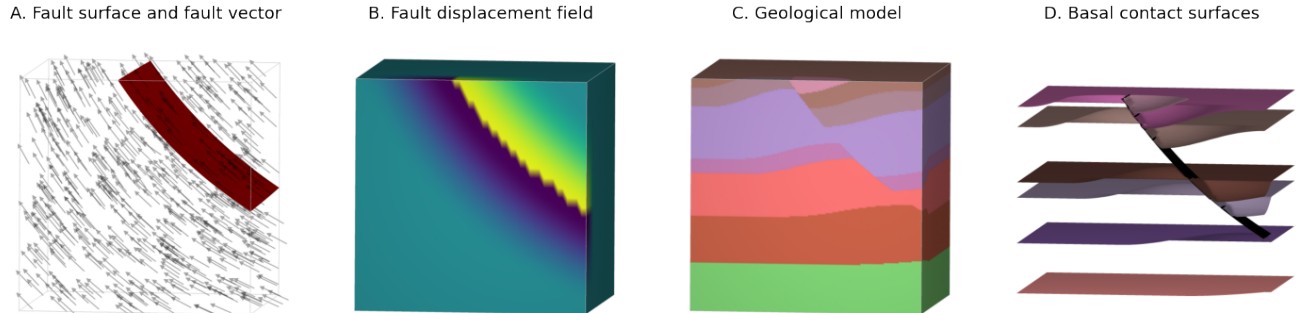

**Figure 6.** Finite fault applied to flat lying stratigraphy. A) Fault displacement vectors and isosurface of fault surface scalar field, B) Scalar field of fault displacement, C) scalar field of stratigraphy and C) surfaces of stratigraphy

contrast many faults end within the model region of interest, and these should be represented as finite features. The fault
displacement can be described using an ellipsoidal fault volume is an ellipsoid fitting the tips of the fault.

Representing faults using a fault volume means that the displacement of the fault needs to vary within the model area. The volumetric displacement can be defined by combining three 1-D displacement functions that describe the fault displacement as a function of the fault frame coordinates. The fault frame coordinates need to be normalised so that they range from -1 to 1 within the fault ellipsoid. The profile of the fault displacement in $f_0$ defines the geometry of the fault throw. The profile of the
fault displacement in $f_1$ defines the magnitude of the displacement along the fault slip direction (*e.g.* how deep does the fault extend for a dip-slip fault) and the fault displacement profile in $f_2$ describes the magnitude of the displacement along the major axis of the fault surface.

In Fig. 6A the fault surface is shown with the slip vectors and the fault extent field value colouring the fault surface. The corresponding fault displacement field are shown in Fig. 6B. In this example the fault has maximal displacement in the centre
of the fault decreasing to the extents of the fault ellipsoid defined by 1-D functions in each of the fault frame coordinates. Different profiles can be chosen either to fit observations or to enforce a particular conceptual model.

The resulting 3D displacement scalar field is shown in Fig. 6A, where displacement scales from 0 to ±1 close to the fault. This displacement field and the fault slip direction vector field can be added to the implicit surface scalar field (shown in Fig. 6C). Fig. 6D shows the faulted stratigraphic surfaces where the implicit function is evaluated using the fault restoration
function. In Fig. 6B and D the displacement of the fault decreases away from the fault and at depth with the displacement not affecting the lowest contact. In Fig. 6C displacement of the fault decreases along the fault with no displacement occurring on the tip of the fault.





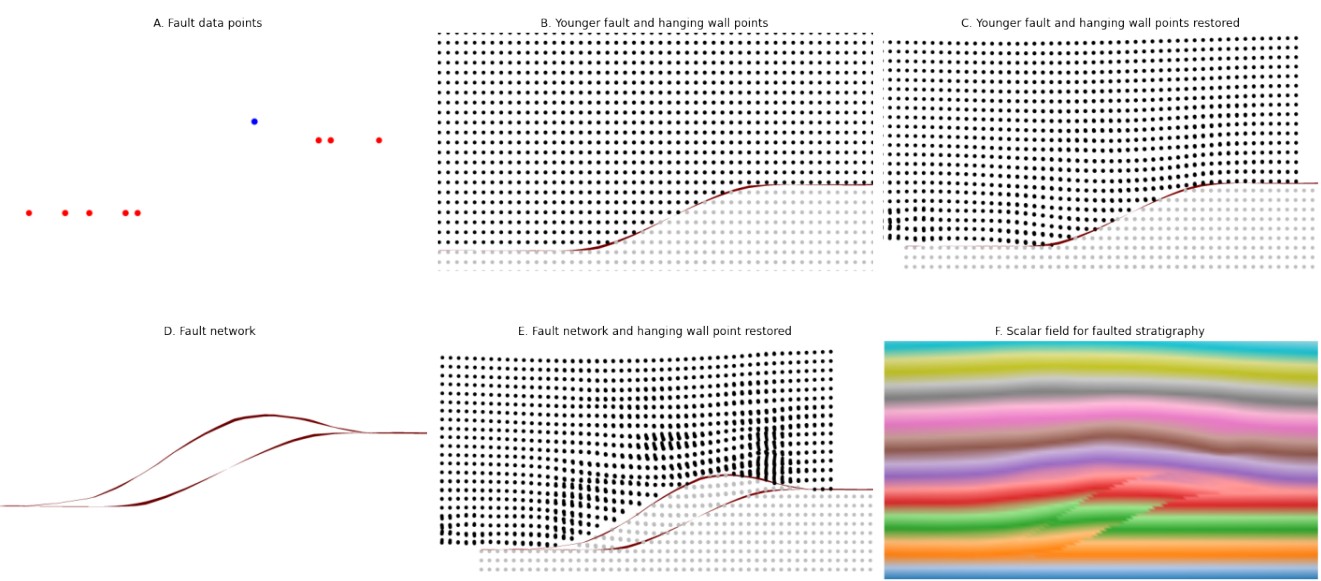

**Figure 7.** Fault duplex applied to flat lying stratigraphy. A) observations of the fault surfaces, red dots showing the location of both surfaces and the blue dot shows the location of the older fault. B) Younger fault surface interpolated with model domain showing hanging wall (black) and footwall (opaque) points. C) hanging wall points restored by younger fault restoration function. D) surfaces for older fault and younger fault. E) model restored by both faults with hanging wall of older fault highlighted by black points. F) duplex fault network added to scalar field of bedding.

## 5.3 Thrust duplex

Fig. 7 is a synthetic case study of a thrust duplex. The observations of the fault surfaces are shown in Fig. 7A, where the red
dots represent the observations of the younger fault location and both blue dot and red dots represent the observations of the older fault location. The fault network shows ramp and flat geometry that is commonly found in compressional settings.

The younger fault is modelled first by interpolating the observations associated with this fault, the resulting surface is shown in Fig. 7B. In Fig. 7B the model domain is highlighted by the black (hanging wall) and opaque points (footwall). The kinematics of the younger fault are applied to the model nodes shown in Fig. 7C where the kinematics are applied in the reverse
and hanging wall nodes have been restored. The younger fault splays from the older fault where the ramp occurs, and both faults share the same geometry at the flat (defined by the red data points). To ensure that the geometry of the interpolated surfaces is the same at the flats the overlapping interpolation nodes from the younger fault are used as equality constraints for the older fault interpolation. The older fault surface observation (blue dot) is restored to its original location and the surface is interpolated combining this observation and the equality constraints from shared geometry of the flats. The younger fault is
added to the implicit function of the older fault and both faults are shown in Fig. 7D. The older fault shows a fault bend fold that is the result of the movement of the younger fault.



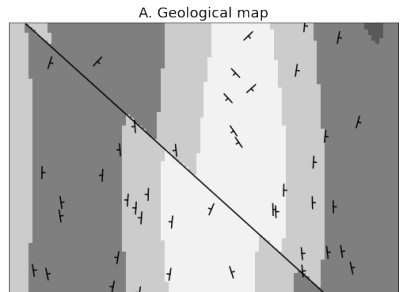 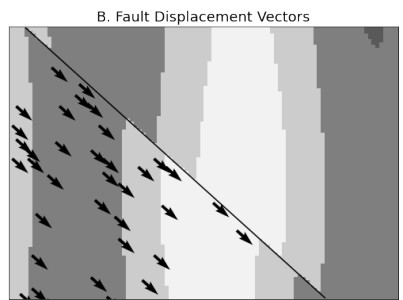 

**Figure 8.** Geological map showing: A) structural observations for bedding, B) Interpolated Fault restoration vectors and C) Restored map

To demonstrate the combined kinematics of this fault network the faults are added to the implicit function representing flat lying stratigraphy Fig. 7F. On the footwall of the younger fault the layers are flat lying and there is no deformation. Between the younger and older fault there is a horse where there is minor rotation of stratigraphy. Above the older fault there is a fault bend fold that can be seen in the stratigraphic layers. These geometrical features are all resulting from the modelled fault kinematics and not mechanics.

### 5.4 Faulted fold series

In the following example we model a faulted doubly plunging fold. The reference model was generated using Noddy (Jessell and Valenta, 1996) and the orientation of the fault surface, fold axial surfaces and folded surface were sampled at irregular locations shown in the geological map (Fig. 8).

To model a folded and refolded structures we use the framework introduced by Laurent et al. (2016) and Grose et al. (2018). To model the faulted fold series the most recent structure needs to be modelled first. In this case this is the fault as it overprints the fold series. We model the fault frame by first interpolating the fault surface using observations of the fault surface (fault trace and fault dip). The fault slip direction is then interpolated so that the gradient of the fault slip field is parallel to the fault slip observations and orthogonal to the gradient of the fault surface. The fault extent can then be interpolated so that its gradient is orthogonal to both the fault slip and the fault surface gradients. The fault surface (red) and fault slip vectors (black arrows) are shown in Fig. 8B. In this example the reference model fault displacement of $1000m$ is used to define the fault displacement.

The fold frame for F1 is interpolated in a similar way to the fault frame, by first interpolating the axial foliation and then interpolating the fold axis direction field (Laurent et al., 2016). The fault is added to the fold frame meaning that the interpolation of the fold frame occurs after the observations have been restored. The scalar fields for the fold frame can be seen in Fig. 10A and B.

Within the fold frame, the intersection lineation is calculated by finding the cross product between the normal to observations of bedding and the gradient of the interpolated axial foliation. The fold axis rotation angle is calculated by finding the angle between the intersection lineation and the fold axis direction field. The fold axis rotation angle is interpolated throughout the model by fitting a Fourier series to the calculated fold rotation angles. We use the Levenberg-Marquardt algorithm for fitting

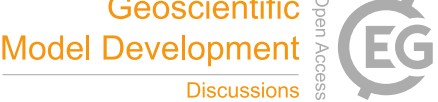

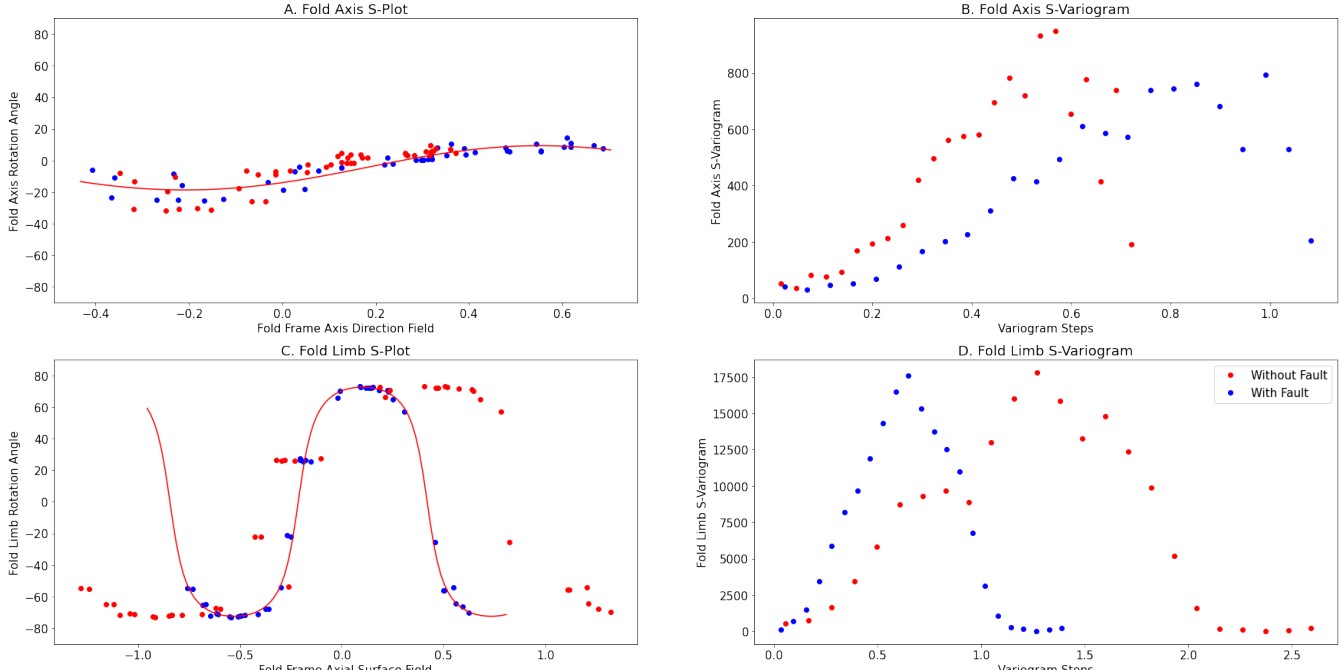

**Figure 9.** Fold geometry plots for the fold axis geometry and fold limb geometry: A. S-Plot showing the variation in fold axis rotation angle with respect to the fold axis direction field showing the fold axis with and without the fault. B. S-Variogram of the fold axis rotation angle. C. S-Plot of the fold limb rotation angle with and without the fault. D. S-Variogram of the fold limb rotation angle

the non-linear least squares problem implemented in the curvefit function found in scipy optimise module. The fold axis can be locally defined by rotating the gradient of the fold axis direction field by the fold axis rotation angle around the gradient of the fold axial foliation. The fold limb rotation angle is calculated by finding the complementary angle to vergence using the interpolated fold axis. Similar to fitting the fold axis rotation angle the fold limb rotation angle is fitted using a Fourier series function and the Levenberg-Maequardt algorithm to find optimal coefficient and wavelength values.

In Fig. 9A. the fold axis rotation angles are plotted against the fold axis direction field coordinate. The fold axis rotation angle is shown (1) where the fault has been added to the interpolator (as blue dots) and; (2) where the fault has not been included (as red dots). This stronger impact can be explained by the fact that the wavelength of the fold limb rotation angle is smaller than that of the fold axis rotation angle. The wavelength of the fold limb rotation angle decreases when removing the effect of the fault, whereas the wavelength of the fold axis rotaton angle increases. This highlights that ignoring the fault could lead to biases when characterising the frequency of the folds. The fold axis rotation angle is marginally affected by the fault, because fold axis variations are relatively small and low-frequency as compared to the fault displacement. This low frequency is highlighted by the absense of a no hole effect in the S-variogram (Fig. 9B. suggesting that there is not a full wavelength of the fold observed in the fold axis. When the fault displacement is applied to the points before fitting the fold axis rotation angle





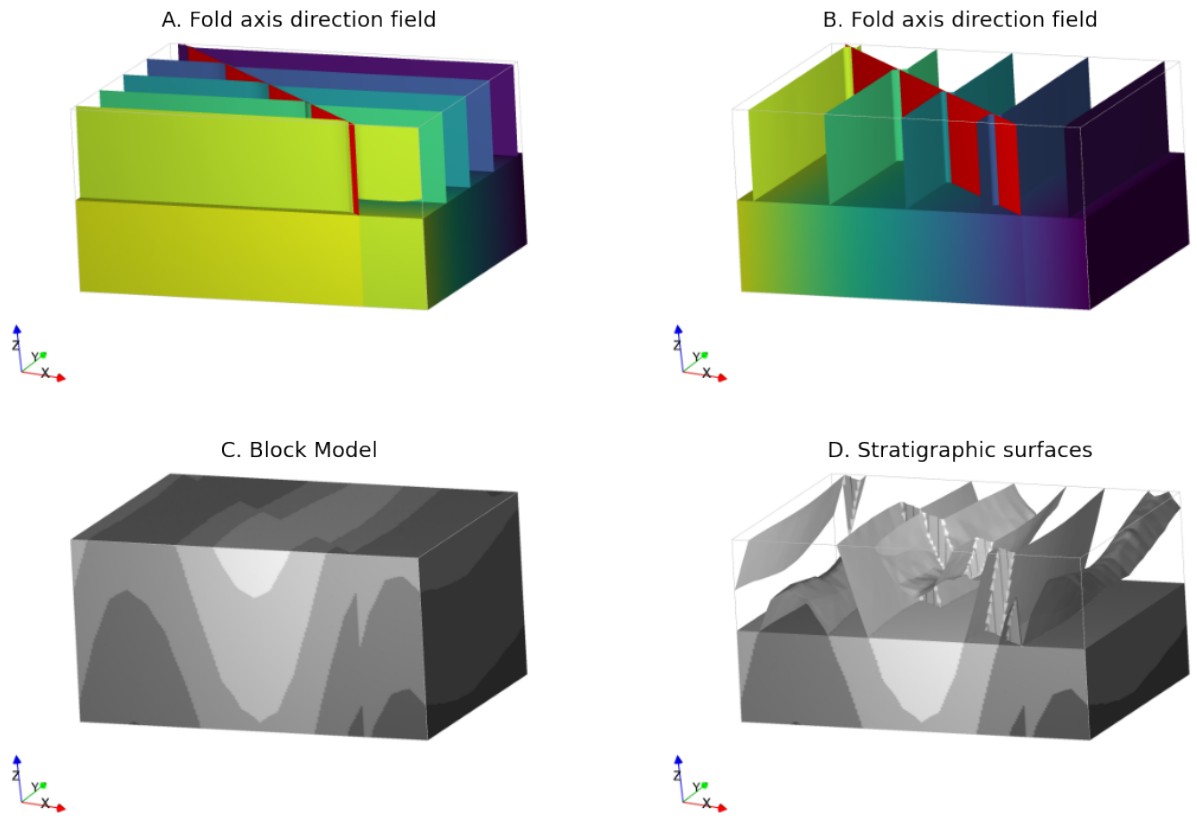

**Figure 10.** Example integrating fault kinematics and fold modelling from Laurent et al. (2016) and Grose et al. (2017). A. Fold frame axial foliation scalar field. B. Fold frame fold axis direction scalar field. Corresponding fold S-Plots are shown in Fig. 9. C. Scalar field showing bedding interpolated using fold constraints with the fault added to the interpolator. D. Surfaces extracted from the geological model in C. including the fault surface.

the wavelength is slightly larger (blue curve in Fig. 9), corresponding to a more consistent alignment of the fold hinges across the fault. The blue curve shows the fold axis rotation angle profile fitted to the blue points.

The fold limb rotation angle is shown in Fig. 9C. In the S-Plot the limbs of the fold are represented by positive and negative clusters and the locations where the rotation angle crosses 0 are the hinge (Grose et al., 2017). There are three limbs observed in the data points, one around an axial foliation value of $-2$, another around an axial foliation value of 0 and a final one around

an axial foliation value of 2. Where the fault is added to the interpolator, the limbs at 0 and 2 are shifted towards the limb at $-2$. In this case, the variation in orientation shown in the S-Plot could either be explained by adding a fault (as done here) or varying the fold wavelength between fold limbs.





# 6 Discussion

A single fault is challenging to incorporate into 3D geological modelling because it defines a discontinuity in the older geo-
logical surfaces. This challenge is further complicated because the fault is not only defined by the location of the discontinuity
but by the kinematics associated with the displacement of the faulted objects. This means that to properly model faults, the
kinematics need to be directly incorporated into the modelling process. These complications are further exaggerated when
modelling a fault network where there are multiple overprinting and interacting faults.

In this contribution faults are modelled using a curvilinear coordinate system and fault operator adapted from Laurent
et al. (2013) and Godefroy et al. (2018) to directly incorporate fault kinematics into the surface description. Our approach
applies the fault operator in reverse prior to interpolating the surface, meaning that the fault description is incorporated into
the implicit surface representation rather than applied to an already discretised surface. Both approaches require building a
reference coordinate system for the fault that defines the fault surface, fault slip direction and the fault extent. The coordinate
system can be easily constrained from observations where the fault slip direction must always be orthogonal to the fault surface
and the fault extent is always orthogonal to the fault slip direction and the fault surface. We show that this coordinate system
can be built allowing for multiple overprinting and interacting faults to be modelled using a serial approach where the scalar
fields representing the coordinates are modelled sequentially. While this approach produces an acceptable coordinate system
for the faulting, it can be costly to evaluate especially on high resolution models. The following strategies could be applied for
modelling large fault networks: (1) Lower the resolution of the interpolation support for the fault network, as the geometry of
the fault should be less complicated than stratigraphy. (2) Only model the fault geometry within the fault volume.

One of the key advantages of our approach is the ability to make use of kinematic observations for constraining the geometry
of faulted geological surfaces. However, in a lot of cases (*e.g.* geological survey maps, potential field geophysical interpreta-
tions), the required kinematic indicators are unknown. One approach for solving this problem would be to build the fault frame
by aligning the fault surface with observations of the fault surface (this step is no different to any other implicit modelling
approach) and second to align the fault extent direction with the map extent of the fault. This defines the fault volume as an
ellipsoid where the major axis is aligned with the map pattern of the fault. The fault slip direction is then orthogonal to both
of these interpolated fields. This approach would provide a good first pass approach for modelling the fault displacement and
is a similar outcome to the inferred slip direction used by step functions, albeit using our approach the fault kinematics are
modelled appropriate to the dip of the fault.

One of the challenges for the presented approach is determining the correct displacement magnitude and profile. Although
in some cases these parameters are observations recorded on a geological map they are not always present. Godefroy et al.
(2018) applies particle swarm optimisation for these parameters by optimising the displacement to the data points within the
fault volume. In a similar approach, an objective function could be applied to the restored continuous surface minimising some
metric, for example for folding this could be incorporated into the probabilistic geological inversion presented by Grose et al.
(2018, 2019). In this framework, fitting a fold model to the S-Plot could be used to determine whether a displacement parameter
and fault profile are acceptable. Fig. 9C shows the fold limb rotation angle for the faulted fold series where the fault has been





added (blue dots) and where the fault hasn't been added (red dots). Adding the fault changes the shape of the S-Plot, and could be added as a likelihood function in an inversion framework. As with all geological problems this would be highly uncertain and require additional domain knowledge to result at a solution.

## 7 Conclusion

The geometry of fault networks and the displacement of faults on other geological surfaces is challenging to integrate into implicit 3D modelling algorithms. Directly integrating faults into the implicit description of surfaces using step functions does not allow for kinematic observations to be incorporated and will result in fault geometries with inconsistent kinematics. In order to mask these flaws, it is possible to constraint fault geometries using interpretive constraints. In this contribution the kinematics are applied in reverse to restore the observation into a pre-fault location. Within the restored model space a continuous surface can be evaluated. The interpolation occurs within the restored model space meaning that no discontinuities or re-meshing are required to generate the models. The restoration kinematics are then added to the implicit description of the model surface allowing for the surface to be evaluated capturing the fault location and the kinematics of the fault. Our approach can be applied to parametric modelling of folds where the fault restoration is applied to the fold geometry characterisation. We demonstrate applications to multiple synthetic examples including a faulted doubly plunging fold series. The flexible description of our model geometries allows more sophisticated kinematic models to be integrated into the work flow.

*Code and data availability.* All of the 3D modelling examples presented in this paper have been generated using the open source 3D modelling package LoopStructural. LoopStructural can be downloaded from github https://www.github.com/Loop3d/LoopStructural or installed using pip install LoopStructural. The examples presented in this paper are included on an Jupyter notebooks including datasets are provided at 10.5281/zenodo.4698304 and can be run using the provided docker file or google colab.

*Author contributions.* All authors contributed to the conceptual design of the fault implementation. LG implemented the methods in LoopStructural. LG prepared the manuscript with help from all authors editing and improving.

*Competing interests.* There are no competing interests

*Acknowledgements.* The authors would like to thank Dr. Gabriel Godefroy for fruitful discussions and reading an early draft of this manuscript. This research has been supported by LP170100985: Loop - Enabling Stochastic 3D Geological Modelling is a OneGeology initiative funded by the Australian Research Council and supported by Monash University, University of Western Australia, Geoscience Australia, the Geological Surveys of Western Australia, Northern Territory, South Australia and New South Wales as well as the Research for



Integrative Numerical Geology, Universite de Lorraine, RWTH Aachen, Geological Survey of Canada, British Geological Survey, Bureau de Recherches Géòlogiques et Minières and Auscope.





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
