# Peer review of "Modelling of faults in LoopStructural 1.0"

_Geoscientific Model Development, 2021_

## Referee Comment (RC2)

How to reconstruct accurately, automatically and without bias the 3-dimension structural architecture of a geological volume based on sparse geological and structural data? Over the past two decades, the authors have developed and incrementally improved a workflow to answer this question. The paper is well structured and makes a commendable effort to guide the reader unfamiliar with the topic and place the new approach they propose within the context of earlier works.

The paper is cognitively challenging, and demands some level of mental flexibility, … I am still chewing on the finite rotation of the gradient of a vector field by the angle this vector field makes with the gradient of another but related vector field, which is perpendicular to the geological structure it represents. Fortunately, the paper includes some examples of application to document the advantage of their new approach, and a Jupyter notebook to allow readers to test and explore the proposed workflow.

The proposed new approach presents some significant advances and for this reason, the paper is definitively worthy of being published. However, the text and the figures demand some attention (see below). Given the difficulties inherent to the subject, the text must be as accurate as possible, and as easy to read as possible. I invite the authors to carefully revisit the text with the objective to improve its readability. The lack of appropriate punctuation must be corrected.

Kind regards,

Patrice

Title: In the title, consider removing "Realistic".

Line 1: … *faulted surface observations*. This is a bit confusion. Replace with "surface offsets" or "marker offsets" as my surface observations are never faulted.

Line 2: Remove "… *that are seen in all tectonic settings*" and enjoy a shorter sentence.

Line 5: Drop a full stop after "… *into geological surface modelling*", and start a new sentence with One approach …

Line 7: Since there are two approaches, replace "*Another approach builds*…" by "The second approach builds …"

Line 11: Add a coma after "*In this study*…". Add a coma after "… *implicit surface* …", add a coma after "… *restore observations*…".

Line 12-13: Replace … *prior to interpolating* … by "prior to interpolate" and consider using comas in this long sentence.

Line 13: Replace "*This approach can build* …" by " This new approach can build…".

Line 14; "… *complex fault stratigraphy* …" ??? consider " … complexly faulted stratigraphy…"

Line 15: Consider replacing "*Our approaches show …*" by "Our approach shows…"

Replace " … *the faulted surface geometry …*" by " … the faulted surface …"

Line 17: Add a coma after "(e.g., intrusions, fold series)" and a coma after "e.g.".

Line 21: Replace "… *from only field observations …*" by "… from field observations only …".

Line 23: We rarely use our biases; we fall victim of them. So, I propose: "Structural geologists typically use their knowledge and expertise to formulate a working hypothesis in a digital format, which often derives from a biased conceptual mental picture (Jessell et al., 2014)."

Line 31: Add a coma after "In this contribution". In general, can you please – when appropriate - add comas to facilitate the reading.

e.g. add a coma at the end of: "*Rather than applying the fault kinematics to the modelled surfaces …*".

Full stop missing at the end of line 44.

Line 47 replace "e.g." by the more appropriate "e.i.*,*" and don't forget the coma after "e.i.".

Line 56: Add coma after "… *model realisations*".

Line 57: Add coma after "… *In this probabilistic framework*".

Line 62: Replace "… *to an already interpolated surface.*" By "… to a surface already interpolated.".

I am not familiar with the concept of "fault ellipsoid". Can you please briefly explain?

Line 83: Remove "*by*" in "… *previous work for modelling folds by …*".

Line 83: Add a coma after "*In this paper …*".

Line 84: Remove "*used*" in "… *for in integrating faults used in LoopStructural …*".

Line: …. *ranging in complexity from … to a thrust …., and to a faulted …*

Line 136 and 144: Replace "*work flow*" by workflow, here and everywhere in the text.

Line 136: Add a coma after "*In their approach …*".

Line 156: "*Trishear has been used to understand the structural evolution that has resulting …*" … replace "*resulting*" by "resulted".

This first part of the paper is certainly interesting and interesting, but the absence of illustration means that readers have only textual information to develop an understanding of the 3D geometrical meaning of each approach. Adding some figures would definitively help readers who not familiar with this topic.

Line 181: "… *when evaluated in the model domain*…" add comas at both ends.

Figure 2: A/ Simple reverse fault ???!!! am I losing my structural geology, or is it a normal fault?

Line 228: Replace all instances of "*work flow*" by "workflow".

Line 268: Replace: "… *the displacement of the fault is separated onto multiple* …" by " … the fault offset is partitioned onto multiple … "

Line 275: Add a coma after: "*To model multiple connected faults* …"

Line 290: Remove the coma (I can't believe I'm making that request) after "…*containing the intersection*…".

Line 291: Replace "*e.g.,*" by "i.e.,".

Line 291: Fix the double "*an*".

Line 294: Add a couple of comas.

For coherence, in figure 3, please use A, B and C, instead of a, b and c.

Line 311: Add a coma after, "*In the following section* …" and consider removing "*will*".

Line 313: Jupyter notebook? Excellent!

Line 321: Replace " … *500m* …" by "… 500 m …".

Figure 5: Red? To me it looks brown?

Line 336: I suggest: "Applying a constant displacement implies making the assumption that …"

Line 340: This sentence doesn't work.

Figure 7D: Can you please use different colours for the young and old faults?

Line 386: "*The fault surface (red)* …" , yet in figure 8, the fault trace is black.

Line 396: "*The fold axis can be locally defined by rotating the gradient of the fold axis direction field by the fold axis rotation angle around the gradient of the fold axial foliation.*" Who am I to argue against that … seriously this is cognitively very demanding, even with the help of figure 9. Consider replacing "*The fold axis can de locally defined …*" by "The orientation of fold axis can de locally obtained …".

Line 410: I don't see any blue curve in Fig.9, and I am not colour blind … I think.

Line 456: … *displacement of faults* … ? Doesn't sound right.

Line 456: Remove "…on other geological surfaces…".

In all figures, please capitalize the first letter of each sentence, e.g., replace " *… offset. B) shows the same …*" by "… offset. B) Shows the same …".

Figure 1: In panels C and D we can't see the fault properly.

Figure 2: Your "Simple reverse fault" looks like a simple reverse fault. Check also the figure caption, which also refers to a reverse fault.

Figure 3: Consider using plural: fixed interpolation nodes, active interpolation nodes, and inactive interpolation nodes. Also, inactive interpolation nodes are represented by white circles in panel c), but the key shows a pale grey circle.

Figure 5: The caption refers to a "red surface", I see none in the figure.

Figure 6: There is a variation in the thickness of rock formations in the footwall. Is this the results of the modelling? Consider "fault vector field" instead of "fault vector". Add a full stop at the end of the figure caption.

Figure 7: In panel D using a different colour for the younger fault. Capitalize … e.g., C) Hanging wall points …

Figure 8: Be consistent … A. Geological map, yet B. Fault Displacement Vectors, and C. Restored map. Either you capitalize or you don't. Same in the figure caption …

Figure 9: Be consistent, and consider replacement in the caption: "*A.*" by "A)", "*B.*" by "B)" etc. To be consistent with other figures.

Figure 10: "*Block Model*", yet "*Stratigraphic surfaces*", be consistent.

---

## Author Response (AR1)

August 23, 2021

**Subject: Response to Reviewers**
Reference: GMD-2021-112

Dear editor and reviewers,

The following major changes were made to the manuscript in response to the reviewers recommendations.

- Title has been changed to 'Modelling of faults in LoopStructural 1.0'.
- Two new figures have been added 1) to help the reader visualise the differences between the different existing methods for 3D modelling faults. 2) showing the fault frame and its relationship to a fault dataset (fault trace and plane geometry).
- Grammatical changes suggested have all been made and manuscript has been rechecked for consistency.
- Figures have been modified to have consistent lower case indexing for subtitles e.g. (a), (b). and capitalisation in the captions. All figures are referenced in text.
- Repository containing notebook examples has been updated and a new Zenodo doi has been generated for the examples and LoopStructural.

Further responses to the reviewers comments are included below and a version of the manuscript highlighting the changes is also included in the submission.

Kind Regards,

Dr Lachlan Grose

Below are the detailed answer to Italos comments:
* * *
Section 3 explains the methodology's core, so I believe it is important to include a figure showing exatcly how f0, f1, and f2 are measured and how the field data is encoded in order to constrain their values (as scalars, displacement vectors, etc.).

▷ We have added a new figure that has three panels. The panel includes a map of a fault, the third shows the fault ellipsoid and the last panel is the fault frame.
* * *
When applying the displacement field, is there any risk of generating "knots", i.e. of a point landing on top of another or past it? In other words, is the relative ordering of the points always maintained?

▷ This is now addressed in 4.2. The displacement is applied iteratively using small step sizes to ensure the order is always maintained.
* * *
It seems it is necessary to label the faults according to their relative age. What if this information is not available?

▷ When modelling faults that do not cover the whole model domain (finite faults), it is only necessary to know the relative timing of faults that are close enough to interact. In general, this should be captured on the geological map in the input data (see Jessell et al., (2021, GMD) for an example of how to extract this from a map). In the cases where the map patterns do not intersect but the faults do at depth then it would be necessary to label the faults. We will add this point into section 3.3.
* * *
Section 4.1: if the coordinates are interpolated sequentially, wouldn't it be possbible to obtain coordinate 2 directly from the cross-product $\nabla f0 \times \nabla f1$?

▷ It is possible to obtain the direction of coordinate 2 but this does not provide the scalar field that measures the distance in that direction from the fault centre. The scalar field is required for defining the fault displacement magnitude. A sentence in 4.1 was added to explain this.
* * *
In Figure 8B the fault displacement vectors appear only on one side of the fault, while in 6A they appear in both sides. Is this correct?

▷ This is intentional, the fault in 8B has a constant displacement on the hanging wall and 0 on the footwall. The fault in 6 has a relative displacement that is -ve on the footwall and positive on the hanging wall to define a "finite fault". We will update the captions to clarify this.

Below are the detailed answer to Patrice's comments, the grammatical corrections and rephrasing have all been implemented and can be seen in the track changes document.
* * *
**Title:** In the title, consider removing "Realistic".

▷ We have removed realistic from the title.
* * *
I am not familiar with the concept of "fault ellipsoid". Can you please briefly explain?

▷ The fault ellipsoid is the volume of non-zero displacement for the fault. This has been added into section describing volumetric fault displacement.
* * *
This first part of the paper is certainly interesting and interesting, but the absence of illustration means that readers have only textual information to develop an understanding of the 3D geometrical meaning of each approach. Adding some figures would definitively help readers who not familiar with this topic.

▷ We have added a figure (now figure 1) demonstrating the differences between the methods.
* * *
**340:** This sentence doesn't work.

▷ The sentence was removed. The following sentence which starts a new paragraph was replaced with 'To model finite faults the fault displacement needs to vary relative to the proximity to the fault.'
* * *
**Figure 7D:** Can you please use different colours for the young and old faults?

▷ Younger fault is blue and older fault is red now. This matches the input data colour scheme.
* * *
**386:** "The fault surface (red) ..." , yet in figure 8, the fault trace is black.

▷ Updated to "The fault trace (black line) .. "
* * *
**396:** "The fold axis can be locally defined by rotating the gradient of the fold axis direction field by the fold axis rotation angle around the gradient of the fold axial foliation." Who am I to argue against that ... seriously this is cognitively very demanding, even with the help of figure 9. Consider replacing "The fold axis can de locally defined ..." by "The orientation of fold axis can de locally obtained ...".

▷ Updated the sentence. The folding method is explained in more details in Grose et al., 2021, Grose et al 2017 and Laurent et al., 2016.
* * *
**410:** I don't see any blue curve in Fig.9, and I am not colour blind ... I think.

▷ It was missing from the figure. A new version of the figure has the blue curve.
* * *
**456:** ... displacement of faults ... ? Doesn't sound right.

▷ Removed
* * *
**456:** Remove "...on other geological surfaces...".

▷ Removed
* * *
**Figure 1:** In panels C and D we can't see the fault properly.

▷ Updated this figure to have thicker lines showing faults.
* * *
**Figure 2:** Your "Simple reverse fault" looks like a simple reverse fault. Check also the figure caption, which also refers to a reverse fault.

▷ You are correct, we have fixed this mistake in text and in the figure.

**Figure 3:** Consider using plural: fixed interpolation nodes, active interpolation nodes, and inactive interpolation nodes. Also, inactive interpolation nodes are represented by white circles in panel c), but the key shows a pale grey circle.

▷ Changed to plurals and the grey circle has been changed to a white filled circle.
* * *
**Figure 5:** The caption refers to a "red surface", I see none in the figure.

▷ The surface is red using matplotlib 'red' colouring.
* * *
**Figure 6:** There is a variation in the thickness of rock formations in the footwall. Is this the results of the modelling? Consider "fault vector field" instead of "fault vector". Add a full stop at the end of the figure caption.

▷ This figure has been recreated and slightly modified. The vectors are now scaled by the displacement magnitude - and only shown where the length of the vector is $> 0$. The title has been changed to fault vector field. The variation in thickness is due to the modelling. As the displacement needs to transition from 0 at the extent of the fault volume to maximal in the centre.

**Figure 7:** In panel D using a different colour for the younger fault. Capitalize … e.g., C) Hanging wall points …

▷ Younger fault is now blue and older fault is red. Caption updated (and others throughout paper updated).
* * *
**Figure 8:** Be consistent … A. Geological map, yet B. Fault Displacement Vectors, and C. Restored map. Either you capitalize or you don't. Same in the figure caption …

▷ Fixed.
* * *
**Figure 9:** Be consistent, and consider replacement in the caption: "A." by "A)", "B." by "B)" etc. To be consistent with other figures.

▷ All captions (and figures) have been updated to use GMD style (a).
* * *
**Figure 10:** "Block Model", yet "Stratigraphic surfaces", be consistent.

▷ This has been corrected for this and other figures.